# Diagnostic accuracy of the aortic dissection detection risk score alone or with D-dimer for acute aortic syndromes: Systematic review and meta-analysis

Sa Ren[1], Munira Essat[1], Abdullah Pandor[1], Steve Goodacre[1]*, Shijie Ren[1], Mark Clowes[1], Paolo Bima[2], Mamoru Toyofuku[3], Rachel McLatchie[4], Eduardo Bossone[5]

1 School of Medicine and Population Health, University of Sheffield, Sheffield, United Kingdom,
2 Department of Medical Science, University of Turin, Turin, Italy, 3 Department of Cardiology, Japanese Red Cross Wakayama Medical Center, Wakayama, Japan, 4 Emergency Department, Royal Infirmary of Edinburgh, Edinburgh, United Kingdom, 5 Department of Public Health, University of Naples "Federico II", Naples, Italy

* s.goodacre@sheffield.ac.uk

**Data Availability Statement:** All relevant data are within the paper and its Supporting Information files.

## Abstract

### Objectives

To evaluate the diagnostic accuracy of the aortic dissection detection risk score (ADD-RS) used alone or in combination with D-dimer for detecting acute aortic syndrome (AAS) in patients presenting with symptoms suggestive of AAS.

### Methods

We searched MEDLINE, EMBASE, and the Cochrane Library from inception to February 2024. Additionally, the reference lists of included studies and other systematic reviews were thoroughly searched. All diagnostic accuracy studies that assessed the use of ADD-RS alone or with D-Dimer for diagnosing AAS compared with a reference standard test (e.g. computer tomographic angiography (CTA), ECG-gated CTA, echocardiography, magnetic resonance angiography, operation, or autopsy) were included. Two reviewers independently selected and extracted data. Risk of bias was appraised using QUADAS-2 tool. Data were synthesised using hierarchical meta-analysis models.

### Results

We selected 13 studies from the 2017 citations identified, including six studies evaluating combinations of ADD-RS alongside D-dimer>500ng/L. Summary sensitivities and specificities (95% credible interval) were: ADD-RS>0 94.6% (90%, 97.5%) and 34.7% (20.7%, 51.2%), ADD-RS>1 43.4% (31.2%, 57.1%) and 89.3% (80.4%, 94.8%); ADD RS>0 or D-Dimer>500ng/L 99.8% (98.7%, 100%) and 21.8% (12.1%, 32.6%); ADD RS>1 or D-Dimer>500ng/L 98.3% (94.9%, 99.5%) and 51.4% (38.7%, 64.1%); ADD RS>1 or ADD RS = 1 with D-dimer>500ng/L 93.1% (87.1%, 96.3%) and 67.1% (54.4%, 77.7%).

**Funding:** AP, SG, ME, MC, ShR all declare funding from the United Kingdom National Institute for Health and Care Research Health Technology Assessment Programme (project number NIHR151853, available from https://www.crd.york.ac.uk/prospero/display_record.php?ID=CRD42022252121). The views expressed in this paper are those of the authors and not necessarily those of the NIHR or the Department of Health and Social Care. Any errors are the responsibility of the authors. The funders had no role in the study design, in the collection, analysis and interpretation of data; in the writing of the manuscript; and in the decision to submit the manuscript for publication.

**Competing interests:** AP, SG, ME, MC, ShR all declare funding from NIHR Health Technology Assessment Grant NIHR151853. There are no other competing interests.

## Conclusions

Combinations of ADD-RS and D-dimer can be used to select patients with suspected AAS for imaging with a range of trade-offs between sensitivity (93.1% to 99.8%) and specificity (21.8% to 67.1%).

## Introduction

Acute aortic syndrome (AAS) is a life-threatening emergency condition affecting the thoracic aorta that includes acute aortic dissection (AAD), intra-mural haematoma, and penetrating ulcer. Computed tomographic angiography (CTA) scanning of the aorta has high sensitivity and specificity for diagnosing AAS, but incurs significant costs and risks of ionising radiation, which may be substantial if CTA is used in a population with low prevalence of AAS.

The aortic dissection detection risk score (ADD-RS) [1] is a clinical score that can estimate the risk of AAS in patients presenting with symptoms suggestive of ASS (see S1 Appendix). The score allocates one point if the patient has a high-risk condition, one point if they have a high-risk symptom, and one point if they have a high-risk examination finding, to give an overall score between zero and three. A threshold of greater than zero or greater than one can then select patients for further investigation. The ADD-RS can be used in a diagnostic pathway to select patients for imaging or alongside the D-dimer blood test to identify low risk patients who could be discharged without imaging.

Previous meta-analysis has shown that ADD-RS greater than zero has sensitivity of 94% (95% confidence interval (CI) 90%-96%) and specificity of 40% (26%-57%) for AAS, with corresponding estimates of 46% (34%–59%) and 91% (79%–96%) for ADD-RS greater than one [2]. The combination of ADD-RS of zero and a D-dimer less than 500ng/L has sensitivity of 99.9% (95% CI, 99.3%-100%) but with uncertain specificity due to heterogeneity (ranging from 3.5% to 43.6%) [3]. The sensitivity of ADD-RS zero or one with D-dimer less than 500ng/L was 98.9% (95% CI, 97.9%-99.9%) with specificity ranging from 30.2% to 62.5%.

Published guidelines make varying recommendations regarding clinical probability scoring and D-dimer in the diagnostic assessment for AAS. American Heart Association/American College of Cardiology guideline [4] suggest that a low ADD-RS and low D-dimer may be a useful strategy to exclude AAS. European Society for Cardiology guidelines [5] suggest that negative D-dimer levels should be considered as ruling out AAS in patients with low clinical probability. Canadian clinical practice guidelines for diagnosing AAS (see S2 Appendix) recommends assessing clinical probability in a similar but not identical way to the ADD-RS and then no further testing if the clinical probability is low, D-dimer if the probability is intermediate, and imaging if the probability is high [6].

We conducted a systematic review and meta-analysis of the ADD-RS at thresholds of greater than zero and greater than one, alone and combination with D-dimer greater than 500ng/L and estimated the accuracy of an alternative strategy combining the ADD-RS with D-dimer based on the Canadian guideline.

## Methods

A systematic review was undertaken in accordance with the general principles recommended in the Preferred Reporting Items for Systematic Reviews and Meta-Analyses of Diagnostic Test Accuracy (PRISMA-DTA) statement [7]. This review was part of a larger Aortic

Syndrome Evidence Synthesis (ASES) project on Diagnostic strategies for suspected acute aortic syndrome (AAS) [8] and was registered on the International Prospective Register of Systematic Reviews (PROSPERO) database (CRD42022252121).

## Eligibility criteria

Prospective or retrospective studies reporting diagnostic accuracy metrics were eligible if they examined ADD-RS alone or ADD-RS in combination with D-Dimer for diagnosing AAS compared with a reference standard test (e.g. a definitive imaging modality such as CTA, ECG-gated CTA, echocardiography, and magnetic resonance angiography or confirmed/ excluded by operation and autopsy). The study population of interest in our review consisted of people (any age) presenting to the ED with symptoms of AAS, including those with new-onset chest, back, or abdominal pain, syncope, or symptoms related to perfusion deficit. Studies including people with AAS following major trauma or as incidental findings were excluded. Case-control designs were also excluded due to the potential for high bias resulting in inaccurate estimates and are not generally representative of a test's accuracy in a clinical setting [9, 10] (a post hoc change).

## Data sources and searches

Potentially relevant studies were identified through searches of several electronic databases including MEDLINE (OvidSP from 1946), EMBASE (OvidSP from 1974), and the Cochrane Library (https://www.cochranelibrary.com). All database searches were conducted from inception to February 2024. The search strategy used free text and thesaurus terms and combined synonyms relating to the topic of interest (e.g. AAS and diagnostic strategies) with diagnostic testing terms (adapted Scottish Intercollegiate Guidelines Network filter for identifying diagnostic studies) [11]. Searches were supplemented by hand-searching the reference lists of all relevant studies (including existing systematic reviews); forward citation searching of relevant articles; contacting key experts in the field; and undertaking targeted searches of the World Wide Web using the Google search engine. No date or language restrictions were applied on any database. Further details on the search strategy can be found in S3 Appendix.

## Study selection

All titles were examined for inclusion by one reviewer (ME) and any citations that clearly did not meet the inclusion criteria (e.g. non-human, unrelated to AAS) were excluded. All abstracts and full text articles were then examined independently by two reviewers (ME and AP). Any disagreements in the selection process were resolved through discussion or if necessary, arbitration by a third reviewer (SG) and included by consensus.

## Data extraction and quality assessment

Data relating to study design, methodological quality and outcomes were extracted by one reviewer (ME) into a standardised data extraction form and independently checked for accuracy by a second (AP). Any discrepancies were resolved through discussion to achieve agreement. Where differences were unresolved, a third reviewer's opinion was sought (SG). Where multiple publications of the same study were identified, data were extracted and reported as a single study. The study team contacted authors of studies reporting ADD-RS alongside D-dimer and invited them to collaborate on the study and share data reporting all potential combinations of the two tests.

The methodological quality of each included study was assessed using the Quality Assessment of Diagnostic Accuracy Studies-2 (QUADAS-2) tool [12]. This instrument evaluates four key domains: patient selection, index test, reference standard, flow and timing. Each domain is assessed in terms of risk of bias and concerns regarding the applicability of the study results (first three domains only). The sub-domains about risk of bias include a number of signaling questions to help guide the overall judgement about whether a study is at high, low, or an unclear (in the event of insufficient data in the publication to answer the corresponding question) risk of bias.

## Data synthesis and analysis

We undertook meta-analysis to estimate the accuracy of the following index tests: ADD-RS>0; ADD-RS>1; ADD-RS>0 or D-dimer>500ng/L; ADD-RS>1 or D-dimer>500ng/L; and a strategy combining the ADD-RS with D-dimer based on the Canadian guideline, which was positive if ADD-RS>1 or ADD-RS = 1 with D-dimer>500ng/L. We also undertook a sensitivity analysis of ADD-RS>0 and ADD-RS>1 limited to the studies that also evaluated D-dimer to determine whether differences between ADD-RS alone and in combination with D-dimer may be due to study selection.

The diagnostic data with multi-threshold were synthesised using a multinomial meta-analysis model proposed by Jones et al. [13]. The multinomial model explicitly quantifies how accuracy depends on the values of threshold, by accounting for within-study correlations between different thresholds and between sensitivity and specificity. A random effects model was used as it takes into account heterogeneity between studies which is generally expected in studies of diagnostic test accuracy [14].

The diagnostic data for strategy based on the Canadian guideline with a single threshold were analysed using a bivariate random effects meta-analysis model [15]. The bivariate model allows for correlation between the sensitivities and specificities within studies. Further details of the statistical models used are provided in S4 Appendix.

All the analyses were conducted using Markov chain Monte Carlo simulations and implemented in the R software environment using JAGs and rjags software package [16]. Convergence to the target posterior distributions was assessed using the Gelman-Rubin convergence statistic [17]. A total 1,000,000 iterations with a burn-in of 100,000 and thinning of 10 were used to estimate the model parameters.

Results were presented as forest plots and receiver operating characteristic (ROC) plots of sensitivity vs 1-specificity. Estimates of sensitivity and specificity with 95% credible intervals (CrI) were plotted individually against each threshold to illustrate the variations among the synthesised studies. 95% prediction intervals (PrI) were also reported, illustrating the between-study heterogeneity and a range of values that might be expected in a future study.

## Patient and public involvement

Two members of the Aortic Dissection Charitable Trust (https://aorticdissectioncharitable trust.org/) joined the ASES project management team and helped to develop the study proposal. SG presented the findings of this review to a webinar of Aortic Dissection Charitable Trust members and sought their feedback on interpretation of the results.

## Results

### Study flow

Fig 1 summarises the process of identifying and selecting relevant literature. Of the 2017 citations identified, 13 studies [18–30] investigating ADD-RS alone or ADD-RS in combination with D-dimer met the inclusion criteria. The majority of the articles were excluded primarily on the basis of an inappropriate target population (patients with AAS or not suspected AAS),

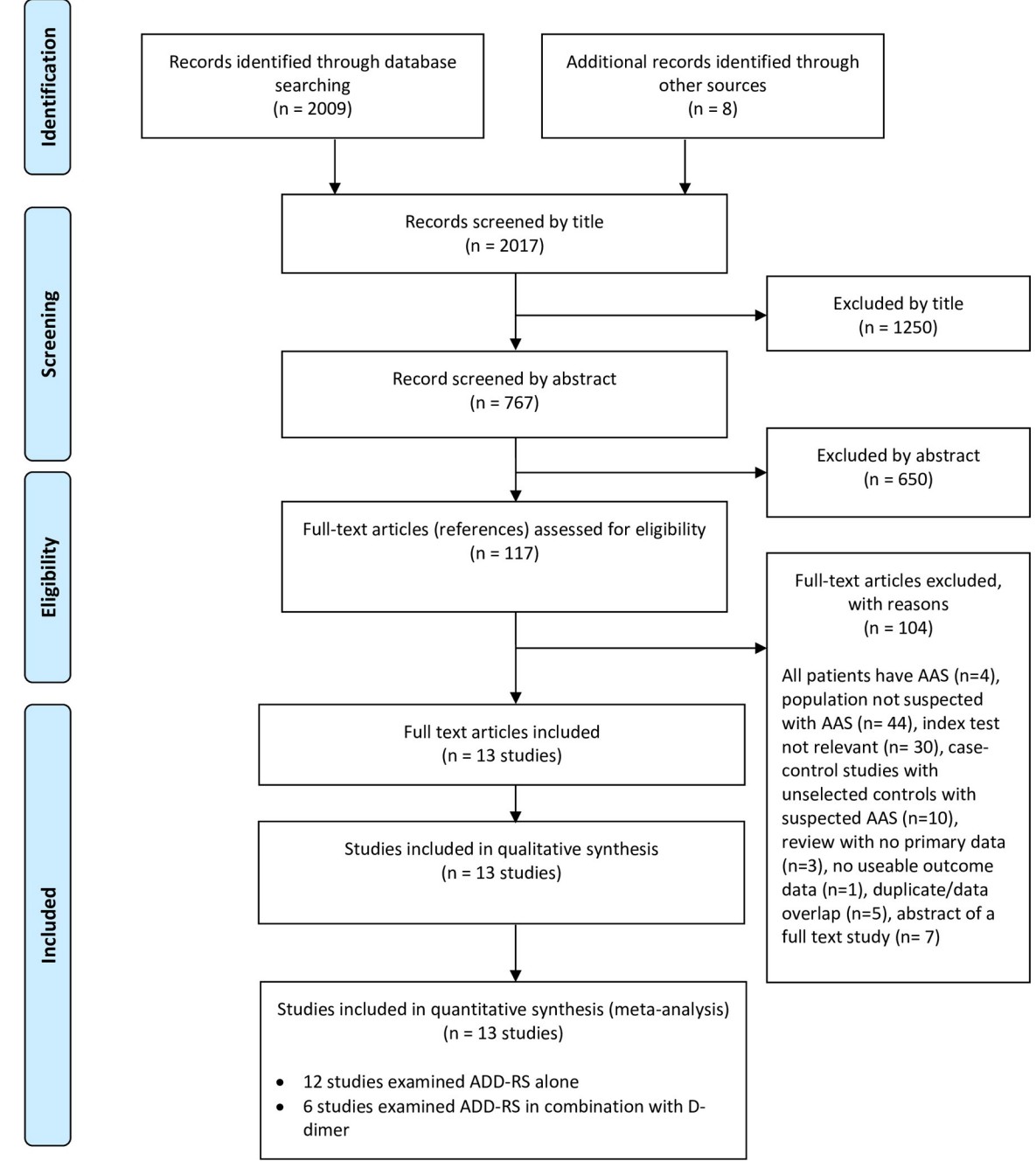

**Fig 1. Study flow chart.**

intervention did not use ADD-RS alone or ADD-RS in combination with D-dimer or an unsuitable publication type (e.g. reviews, or abstract of full text studies). A full list of excluded studies with reasons for exclusion can be found in S5 Appendix. We identified seven studies [20–24, 26, 27] reporting ADD-RS in combination with D-dimer, with six studies [20, 22–24, 26, 27] using a consistent threshold for D-dimer (>500ng/L) and reporting or sharing data allowing all combinations of ADD-RS and D-dimer >500ng/L to be analysed. Two studies [25, 26] reported evaluation of ADD-RS alone and in combination with D-dimer respectively using overlapping data. Meta-analysis therefore included 12 studies of ADD-RS alone and six studies of ADD-RS with D-dimer.

## Study and patient characteristics

The design and patient characteristics of the 13 included studies [18–30] are summarised in Table 1. We report the characteristics of the whole study cohort in this table. In some studies reporting ADD-RS and D-dimer, the analysis of each individual test and the combination involved a subsample that received the relevant test or combination, rather than the whole cohort. Sample size ranged from N = 162 to N = 22075, with prevalence of AAS ranging from 0.3% to 64%. The two largest studies (Yamashita et al. [30] and McLatchie et al. [23]) reported the lowest prevalence of AAS. The studies with higher prevalence tended to restrict recruitment to those receiving imaging for AAS.

## Risk of bias and applicability assessment

The overall methodological quality of the 13 included studies is summarised in Table 2 and Fig 2 (also see S6 Appendix). The methodological quality of the included studies was variable, with most studies having low or unclear risk of bias and applicability concerns in at least one item of the QUADAS-2 tool. Risk of bias in patient selection was rated as low for studies reporting consecutive sampling and high for studies reporting convenience sampling. However, variation in the definition of the eligible population made judgements about patient selection difficult and may have influenced other quality criteria. The studies of McLatchie et al. [23] and Yamashita et al. [30] appeared to have much more inclusive eligibility criteria but were rated as having high risk of bias in flow and timing, principally due to a substantial portion of patients not receiving a reference test (imaging or follow-up) [23, 30]. Six studies [18–21, 23, 30] had at least one unclear risk of bias in the domain of index test (ADD-RS alone or with D-dimer) or the reference standard due to a lack of clarity as to whether the reference standard results were interpreted without knowledge of the index test or vice versa. Although no studies had high applicability concerns, two studies [20, 21] were considered to have unclear concerns as details of the reference standard tests were not clearly specified.

## Diagnostic performance of strategies

The pooled accuracy estimates for sensitivity and specificity are summarised in Table 3. 12 studies contributed to the meta-analysis for ADD-RS alone; six studies contributed to the meta-analysis for ADD-RS in combination with D-Dimer.

Our meta-analysis has shown that ADD-RS>0 has high sensitivity 94.6% (95% CrI: 90% to 97.5%) and low specificity 34.7% (95% CrI: 20.7% to 51.2%) while ADD-RS>1 has low sensitivity 43.4% (95% CrI: 31.2% to 57.1%) and high specificity 89.3% (95% CrI: 80.4% to 94.8%). Combinations of ADD-RS with D-dimer provide a range of trade-offs between sensitivity and specificity, varying from 99.8% (95% CrI: 98.7% to 100%) sensitivity and 21.8% (95% CrI: 12.1% to 32.6%) specificity for the combination of ADD-RS>0 or D-dimer>500ng/L to sensitivity 93.1% (95% CrI: 87.1% to 96.3%) and specificity 67.1% (95% CrI: 54.4% to 77.7%) for a

**Table 1. Study and population characteristics of the included studies.**

| Author, year | Country/ (centres) | Study design | Population | Sample size (N) | Mean Age | Female (%) | AAS or AAD (%) | Index Test and Threshold | Reference Standard |
|---|---|---|---|---|---|---|---|---|---|
| Chun & Siu, 2023 [18] | Hong Kong (2) | RC | Suspected AAD | 534 | 65.2 | 39% | 37.1% | • ADD-RS (>0; >1) | CTA |
| Deng et al., 2023 [19] | China (2) | PC | Suspected AAS | 200 | 65.2 | 32% | 14.5% | • ADD-RS (>0; >1) | CTA, TEE, MRA, surgery or autopsy |
| Gorla, 2017a [20] | Germany (1) | RC | Chest pain with suspected AAS | 376 | 63.1 | 38.6% | 22.6% | • ADD-RS (>0; >1)<br>• ADD-RS or D-dimer (>0; >1 or 500 ng/ml)<br>• ADD-RS or D-dimer (>1 or 1 and 500 ng/ml) [a] | Imaging |
| Kodera, 2016 [21] | Japan (1) | RC | Suspected AAD | 162 | NR | NR | 64.0% | • ADD-RS (>0, >1)<br>• ADD-RS or D-dimer (>0; >1 or 1000 ng/ml) | Unspecified |
| Kotani, 2017 [22] | Japan (1) | RC | Chest pain with suspected AAS | 887 | 70 | 32.4% | 13.9% | • ADD-RS (>0, >1)<br>• ADD-RS or D-dimer (>0; >1 or 500 ng/ml) [b]<br>• ADD-RS or D-dimer (>1 or 1 and 500 ng/ml) [a] | CT scan |
| McLatchie, 2023 [23] | UK (27) | PC | Patients with symptoms potentially attributable to AAS | 5548 | 55 | 53.3% | 0.3% | • ADD-RS (>0; >1)<br>• ADD-RS or D-dimer (>0; >1 or 500 ng/ml)<br>• ADD-RS or D-dimer (>1 or 1 and 500 ng/ml) [a] | CTA |
| Morello, 2021 [24] | Italy (2) | PC | Suspected AAS | 443 | NR (median 63) | 33.3% | 11.1% | • ADD-RS (>1)<br>• ADD-RS or D-dimer (>1 or 500 ng/ml)<br>• ADD-RS or D-dimer (>1 or 1 and 500 ng/ml) [a] | CTA, TEE, MRA, surgery or autopsy |
| Nazerian 2014a [25] | Italy (2) | RC [c] | Chest/ back/ abdominal pain, syncope or perfusion deficit with suspected AAD | 1328 | NR | 33.6% | 21.9% | • ADD-RS (>0; >1) | CTA or TEE |
| Nazerian 2014b [26] | Italy (2) | RC [c] | Chest/ back/ abdominal pain, syncope or perfusion deficit with suspected AAD | 1035 | NR | 35.4% | 22.5% | • ADD-RS or D-dimer (>0; >1 or 500 ng/ml)<br>• ADD-RS or D-dimer (>1 or 1 and 500 ng/ml) [a] | CTA |

(*Continued*)

**Table 1.** (Continued)

| Author, year | Country/ (centres) | Study design | Population | Sample size (N) | Mean Age | Female (%) | AAS or AAD (%) | Index Test and Threshold | Reference Standard |
|---|---|---|---|---|---|---|---|---|---|
| Nazerian, 2018 [27] | Italy, Switzerland, Brazil, Germany (6) | PC | Chest/ back/ abdominal pain, syncope, or perfusion deficit with suspected AAD | 1848 | 62 | 37.7% | 13.0% | • ADD-RS (>0; >1) • ADD-RS or D-dimer (>0; >1 or 500 ng/ml) • ADD-RS or D-dimer (>1 or 1 and 500 ng/ml) [a] | CTA, TEE, MRA, surgery or autopsy; or 14-day clinical follow-up |
| Ohle, 2019 [28] | Canada, America (2) | RC | Suspected AAS | 370 | NR | 46.5% | 3.2% | • ADD-RS (>0; >1) | CTA |
| Rotella, 2018 [29] | Australia (1) | RC | Suspected AAS | 200 | NR | 47% | 2.5% | • ADD-RS (>0; >1) | CTA |
| Yamashita, 2018 [30] | Japan (44) | RC | Chest/ back/ abdominal pain, syncope, or perfusion deficit with suspected AAD | 22075 | NR | NR | 1.6% | • ADD-RS (>0; >1) | CT followed by radiologist assessment and/or surgical finding |

AAS, acute aortic syndrome; AAD, acute aortic dissection; ADD-RS, aortic dissection detection risk score (ADD-RS); CT, computed tomography; CTA, computed tomography angiogram; MRA, magnetic resonance angiography; TEE, transoesophageal echocardiography; PC, prospective cohort study; RC, retrospective cohort study; NR, not reported.

[a] Strategy based on Canadian rule (data extracted or received from authors allowing any combination of ADD-RS with D-dimer to be evaluated)

[b] Age adjusted

[c] Retrospective analysis of a prospective registry

**Table 2. QUADAS-2 quality assessment summary - Review authors' judgements.**

| Author, year | Risk of bias | | | | | Applicability concerns | | | |
|---|---|---|---|---|---|---|---|---|---|
| | Patient selection | Index test | | Reference standard | Flow and timing | Patient selection | Index test | | Reference standard |
| | | ADD-RS | ADD-RS with D-dimer | | | | ADD-RS | ADD-RS with D-dimer | |
| Chun & Siu, 2023 [18] | High | Unclear | - | Unclear | Unclear | Low | Low | - | Low |
| Deng et al., 2023 [19] | Low | Unclear | - | Unclear | Unclear | Low | Low | - | Low |
| Gorla, 2017a [20] | High | Low | Low | Unclear | Unclear | Low | Low | Low | Unclear |
| Kodera, 2016 [21] | Unclear | Low | Low | Unclear | Unclear | Low | Low | Low | Unclear |
| Kotani, 2017 [22] | High | Low | Low | Low | Unclear | Low | Low | Low | Low |
| McLatchie, 2023 [23] | High | Unclear | Unclear | Low | High | Low | Low | Low | Low |
| Morello, 2021 [24] | Low | Low | Low | Low | Unclear | Low | Low | Low | Low |
| Nazerian 2014a [25] | Low | Low | - | Low | Low | Low | Low | - | Low |
| Nazerian 2014b [26] | Low | Low | Low | Low | Low | Low | Low | Low | Low |
| Nazerian, 2018 [27] | Low | Low | Low | Low | Unclear | Low | Low | Low | Low |
| Ohle, 2019 [28] | Unclear | Low | - | Low | Unclear | Low | Low | - | Low |
| Rotella, 2018 [29] | Unclear | Low | - | Low | Low | Low | Low | - | Low |
| Yamashita, 2018 [30] | Low | Unclear | - | Unclear | High | Low | Low | - | Low |

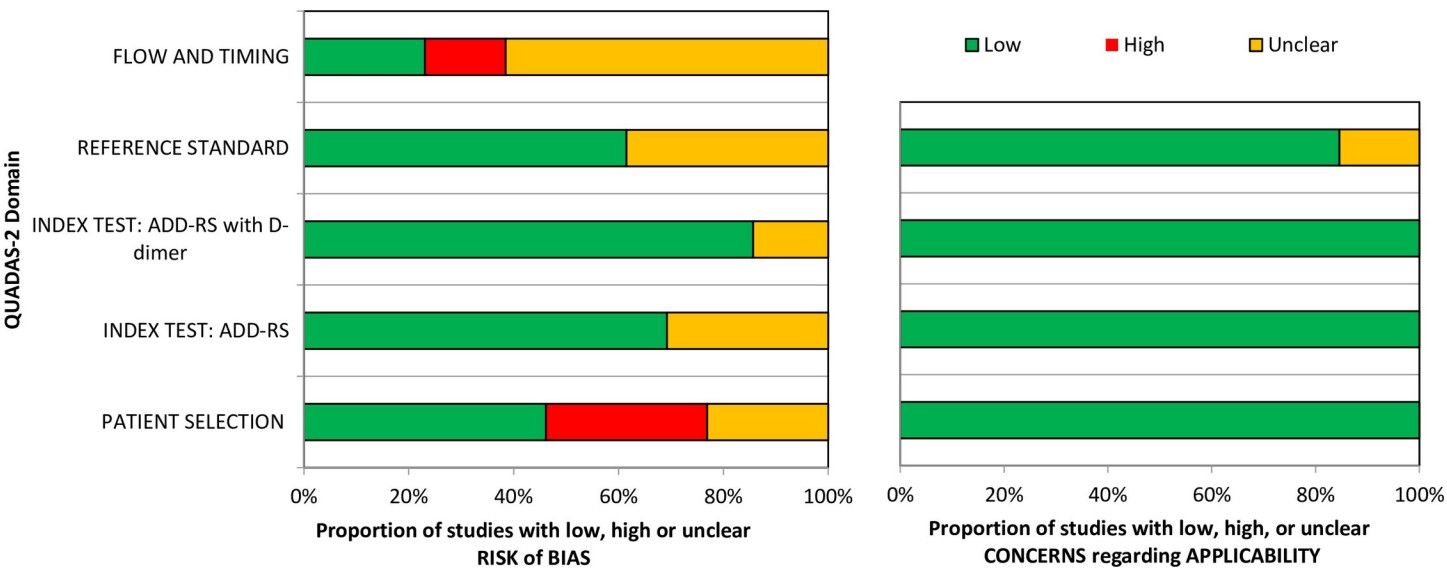

**Fig 2. QUADAS-2 assessment summary graph - Review authors' judgements.**

combination based on the Canadian guideline. The results for these analyses are shown separately in Figs 3 and 4. The forest plots are presented in S7 Appendix. To facilitate the comparison among different strategies, the pooled estimates of each strategy along with their 95% credible intervals are plotted in Fig 5.

**Table 3. Pooled estimates for each strategy.**

| Strategy | Threshold | N | Sensitivity (95% CrI) [95% PrI] | Specificity (95% CrI) [95% PrI] |
|---|---|---|---|---|
| ADD-RS main analysis | ADD-RS>0 | 12 | 94.6% (90%, 97.5%) [72.6%, 99.7%] | 34.7% (20.7%, 51.2%) [3.3%, 86.9%] |
| | ADD-RS>1 | 12 | 43.4% (31.2%, 57.1%) [9.9%, 83.3%] | 89.3% (80.4%, 94.8%) [41.9%, 99.5%] |
| ADD-RS with D-dimer analysis | ADD RS>0 or D-Dimer>500 | 6 | 99.8% (98.7%, 100%) [96.1%, 100%] | 21.8% (12.1%, 32.6%) [2.6%, 50.7%] |
| | ADD RS>1 or D-Dimer>500 | 6 | 98.3% (94.9%, 99.5%) [86.4%, 100%] | 51.4% (38.7%, 64.1%) [18.5%, 83.5%] |
| | ADD RS>1, ADD RS = 1 and D-dimer>500* | 6 | 93.1% (87.1%, 96.3%) [74.1%, 98.3%] | 67.1% (54.4%, 77.7%) [33.4%, 89.3%] |
| ADD-RS sensitivity analysis | ADD-RS>0 | 6 | 95.1% (88.5%, 98.4%) [72.9%, 99.8%] | 38% (20.1%, 59.1%) [4.5%, 86.8%] |
| | ADD-RS>1 | 6 | 41.6% (24.8%, 59.1%) [8.1%, 82.5%] | 91.7% (81.7%, 97%) [53.7%, 99.6%] |

N: number of studies; ADD-RS: Aortic Dissection Detection Risk Score; CrI: Credible Intervals; PrI: Prediction Intervals.
*Canadian guideline.

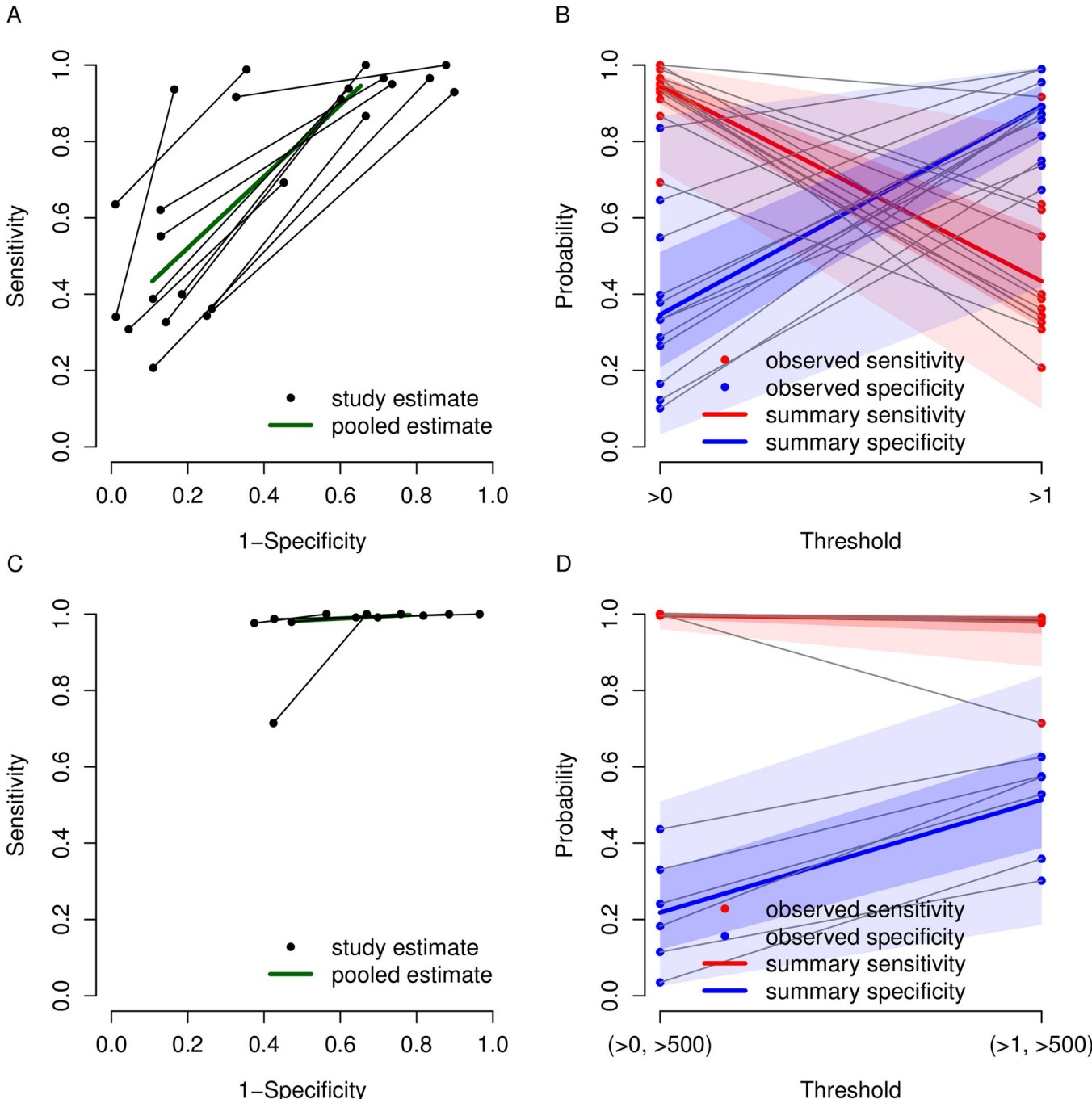

**Fig 3. Summary plots for ADD-RS and ADD-RS in combination with D-Dimer.** The summary plots for ADD-RS alone are presented in panel A and B, and the summary plots for ADD-RS with D-Dimer are presented in panel C and D. Individual sensitivity and specificity from the same study are linked with lines. In panel A and C, receiver operating characteristic (ROC) plots are displayed. In panel B and D, pooled sensitivity and specificity along with the 95% credible intervals and 95% prediction intervals are plotted. The 95% credible intervals are marked by the shaded areas around the summary estimates, showing the range of likely values for average diagnostic accuracy. The 95% prediction intervals are marked by the wider and lighter shaded areas around the summary estimates, indicating a range of values that might be expected in a future study.

95% prediction intervals are used to present the extent of between study variation. Wider prediction intervals suggest larger amount of between study variation. ADD-RS alone has larger amount of between study variations compared to combinations of ADD-RS with

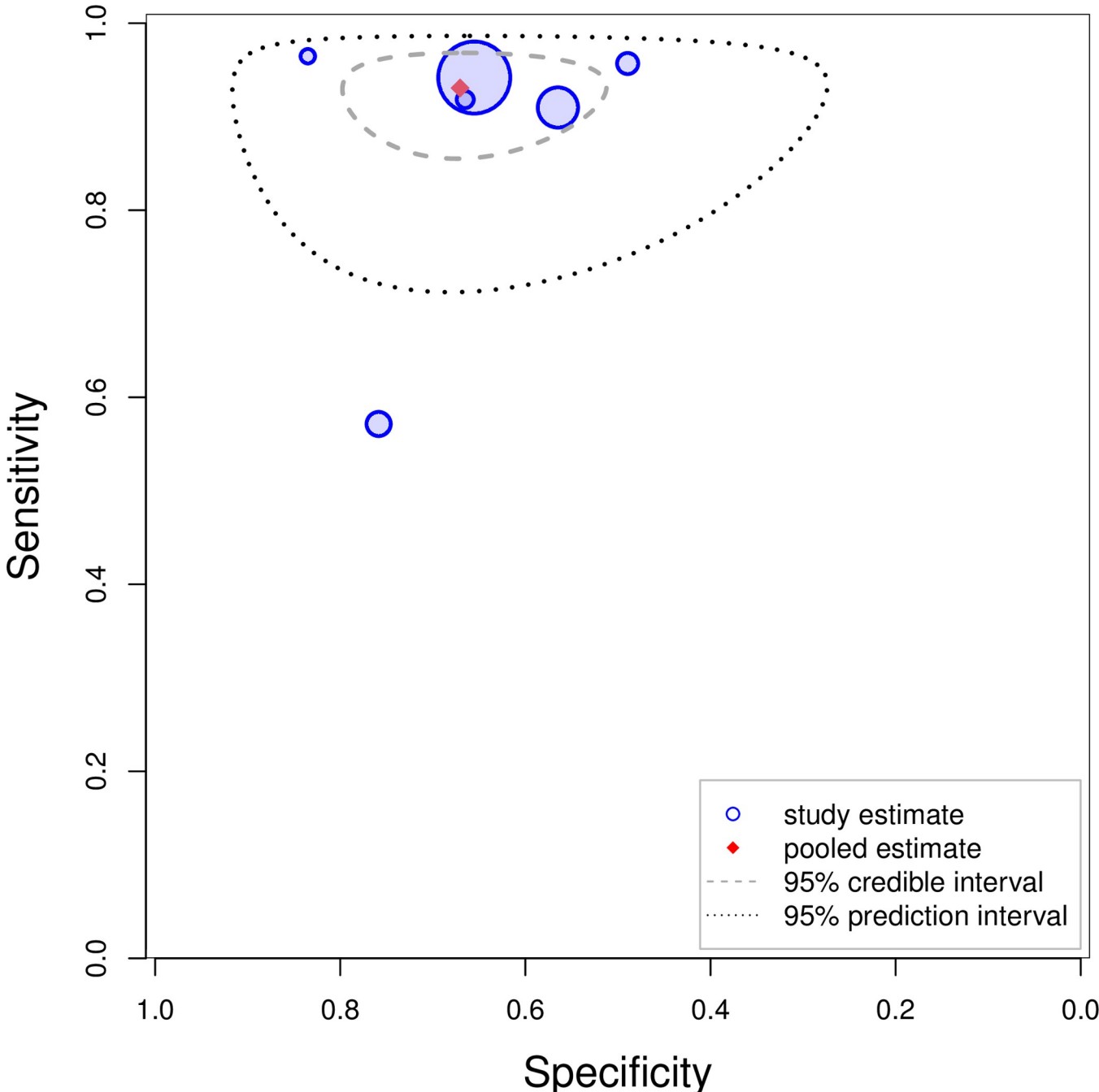

**Fig 4. Summary plot for Canadian guideline.** Blue circles indicate individual studies scaled with the sample size.

D-Dimer for both sensitivity and specificity. ADD-RS>0 or D-dimer>500ng/L (95% PrI: 96.1% to 100%) and ADD-RS>1 or D-dimer>500ng/L (95% PrI: 86.4% to 100%) have narrow prediction intervals for sensitivity, suggesting that there is a small amount of between study variation between sensitivity. For the Canadian guideline, the between-study variance was estimated to be 0.36 (95% CrI: 0.07 to 1.29) for logit sensitivity and 0.40 (95% CrI: 0.20 to 0.79) for

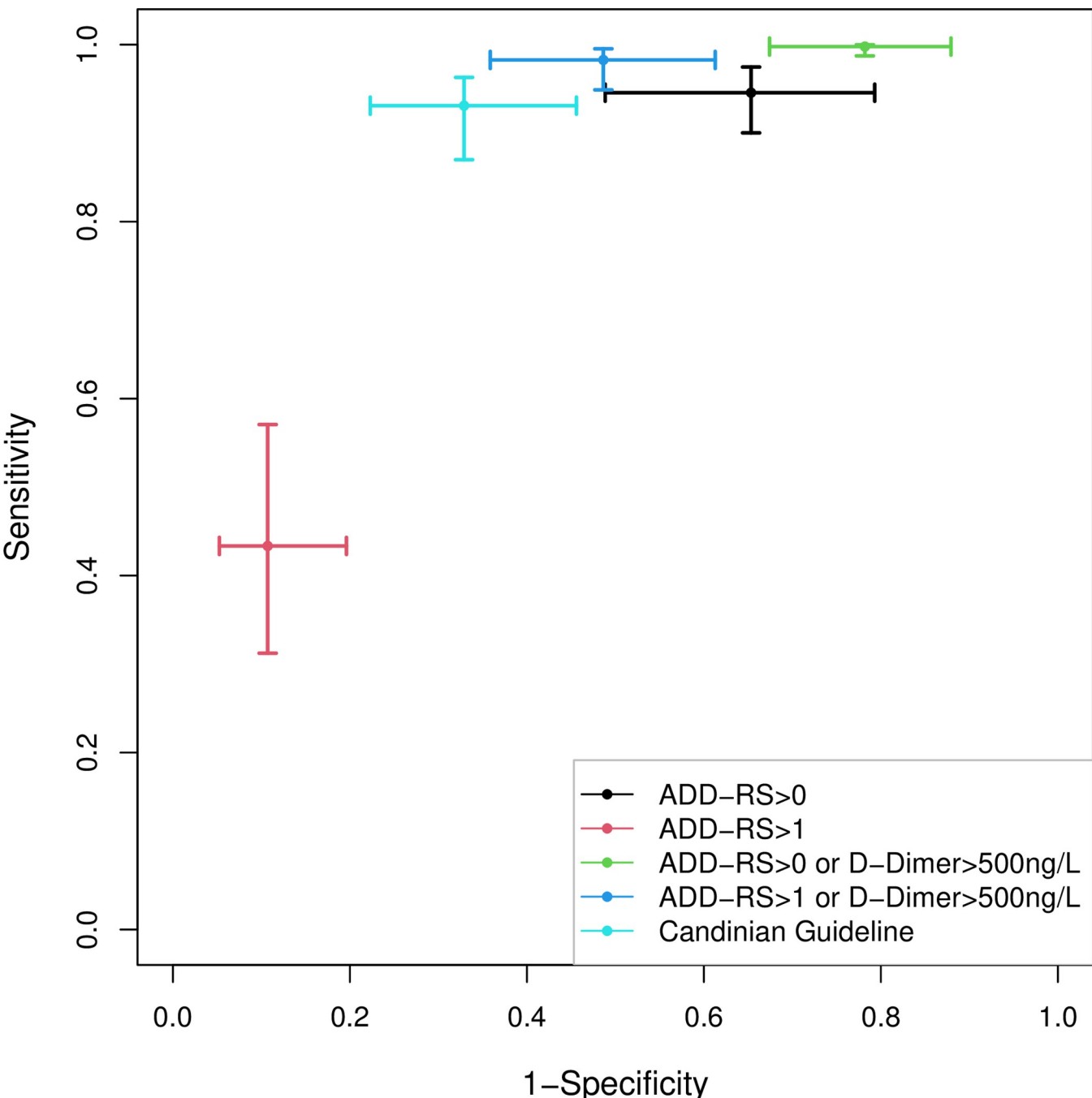

**Fig 5. Pooled estimates and 95% credible intervals for each strategy.**

logit specificity. The correlation coefficient between logit sensitivity and logit specificity was estimated to be -0.03 (95% CrI: -0.78, 0.85).

A sensitivity analysis of ADD-RS>0 and ADD-RS>1 limited to the studies that also evaluated D-dimer was conducted. Six studies contributed to this analysis and the pooled sensitivity and specificity were similar to the ADD-RS main analysis with 12 studies (ADD-RS>0: sensitivity 95.1% vs 94.6%; specificity 38.0% vs 34.7%. ADD-RS>1: sensitivity 41.6% vs 43.4%;

specificity 91.7% vs 89.3%). The most notable difference was that the specificity of AD-RS>0 was higher in the sensitivity analysis because the studies of ADD-RS alone tended to have lower specificity at the ADD-RS>0 threshold. Summary plots and forest plots for the sensitivity analyses are presented in S7 Appendix.

## Discussion

Our analysis shows that strategies using ADD-RS and/or D-dimer have a range of sensitivities and specificities for AAS. Higher sensitivity was generally achieved at the expense of lower specificity. The exception was that the combination of ADD-RS>1 or D-Dimer>500ng/L had superior sensitivity and specificity to ADD-RS>0 alone, when assessed using point estimates. The sensitivity analysis suggested that there was little difference in estimates of ADD-RS accuracy between the analysis including all 12 studies and the analysis limited to the six studies also evaluating D-dimer.

These findings suggest that the combination of ADD-RS and D-dimer can be used to select patients for imaging but the choice of strategy for combining the tests will depend upon whether clinicians prioritise avoiding missed diagnosis or avoiding over-investigation. This could be determined at an individual patient level by exploring patient values and preferences, or at a health service level through using decision analysis to compare the costs and outcomes of alternative strategies. For individual patients, we could also consider the potential risk of false negative assessments in patients with early presentation or longer lasting symptoms.

Our summary estimates are similar to those in previous meta-analyses [2, 3], albeit with slightly lower sensitivity. The novel element of our analysis in inclusion of a strategy based on the Canadian clinical practice guideline, which has lower sensitivity but higher specificity than other strategies combining the ADD-RS with D-dimer. This offers an alternative strategy to patients who wish to avoid over-investigation and populations with a low prevalence of AAS, where a strategy with low specificity would generate an unacceptably low yield of positive imaging.

Our meta-analysis used data from six studies [20, 22–24, 26, 27] to estimate the accuracy of a variety of different strategies, including one based on the Canadian clinical practice guideline that has not been evaluated in previous meta-analysis. We were not able to obtain data from a seventh study [21] that evaluated the combination of ADD-RS with D-dimer but used a different D-dimer threshold. There was potentially important heterogeneity between the studies, especially in estimates of specificity, which increases the uncertainty around these estimates. This heterogeneity may reflect differences in study design, particularly patient selection, with studies varying between those reporting populations with a low rate of imaging for the reference standard [23] to those reporting populations with a higher rate of imaging [20, 22, 24, 26, 27].

The combination of ADD-RS and D-dimer can be used to select patients with suspected AAS for further investigation, but the choice of strategy involves a trade-off between sensitivity and specificity. The consequences of missed AAS are potentially catastrophic, so high sensitivity is clearly important, but applying a strategy with modest specificity to a population with a low prevalence of ASS will result in a low yield of positive imaging that may not justify the costs and harms of imaging. Decision-analytic modelling is required to examine the trade-off between sensitivity and specificity, predict the costs and outcomes expected with alternative strategies, and estimate the cost-effectiveness of alternative strategies. Research is also required to determine how strategies combining ADD-RS with D-dimer are used in practice to influence decision-making.

## Conclusions

Combinations of ADD-RS and D-dimer can be used to select patients with suspected AAS for imaging with a range of trade-offs between sensitivity (93.1% to 99.8%) and specificity (21.8% to 67.1%). Further research is required to examine this trade-off and determine the most effective and cost-effective strategy.

## Supporting information

**S1 Appendix. The Aortic Dissection Detection Risk Score (ADD-RS).**
(DOCX)

**S2 Appendix. Diagnostic pathway for patients presenting with clinical features suggestive of AAS (Canadian guideline).**
(DOCX)

**S3 Appendix. Literature search strategies.**
(DOCX)

**S4 Appendix. Statistical models for the meta-analysis.**
(DOCX)

**S5 Appendix. List of excluded studies with rationale.**
(DOCX)

**S6 Appendix. QUADAS-2 quality assessment.**
(DOCX)

**S7 Appendix. Summary plots and forest plots for meta-analyses.**
(DOCX)

**S1 Checklist. PRISMS DTA checklist.**
(DOC)

## Acknowledgments

The authors would like to thank all additional members of the core project group for NIHR151853 for input and commentary throughout the work. We are also indebted to Joanne Hinde for assistance with logistics and administration.

## Author Contributions

**Conceptualization:** Abdullah Pandor, Steve Goodacre, Shijie Ren.

**Data curation:** Sa Ren, Munira Essat, Mark Clowes, Paolo Bima, Mamoru Toyofuku, Rachel McLatchie, Eduardo Bossone.

**Formal analysis:** Sa Ren, Munira Essat, Abdullah Pandor, Shijie Ren.

**Funding acquisition:** Steve Goodacre.

**Investigation:** Munira Essat, Mark Clowes.

**Methodology:** Sa Ren, Shijie Ren.

**Resources:** Steve Goodacre.

**Visualization:** Sa Ren.

**Writing – original draft:** Sa Ren, Abdullah Pandor, Steve Goodacre.

**Writing – review & editing:** Sa Ren, Munira Essat, Abdullah Pandor, Steve Goodacre, Shijie Ren, Mark Clowes, Paolo Bima, Mamoru Toyofuku, Rachel McLatchie, Eduardo Bossone.

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
