## [Decision Letter · Decision Letter 0]

9 Feb 2024

PONE-D-23-42764Diagnostic accuracy of the aortic dissection detection risk score alone or with D-dimer for acute aortic syndromes: Systematic review and meta-analysisPLOS ONE

Dear Dr. Ren,

Thank you for submitting your manuscript to PLOS ONE. After careful consideration, we feel that it has merit but does not fully meet PLOS ONE’s publication criteria as it currently stands. Therefore, we invite you to submit a revised version of the manuscript that addresses the points raised during the review process.

We look forward to receiving your revised manuscript.

Kind regards,

Amir Hossein Behnoush

Academic Editor

PLOS ONE

“The authors would like to thank all additional members of the core project group for NIHR151853 for input and commentary throughout the work. We are also indebted to Joanne Hinde for assistance with logistics and administration.”

“AP, SG, ME, MC, ShR all declare funding from the United Kingdom National Institute for Health and Care Research Health Technology Assessment Programme (project number NIHR151853, available from https://www.crd.york.ac.uk/prospero/display_record.php?ID=CRD42022252121). The views expressed in this paper are those of the authors and not necessarily those of the NIHR or the Department of Health and Social Care. Any errors are the responsibility of the authors. The funders had no role in the study design, in the collection, analysis and interpretation of data; in the writing of the manuscript; and in the decision to submit the manuscript for publication.”

Reviewers' comments:

Reviewer's Responses to Questions

**Comments to the Author**

1. Is the manuscript technically sound, and do the data support the conclusions?

Reviewer #1: Yes

Reviewer #2: Yes

2. Has the statistical analysis been performed appropriately and rigorously? 

Reviewer #1: Yes

Reviewer #2: Yes

3. Have the authors made all data underlying the findings in their manuscript fully available?

Reviewer #1: Yes

Reviewer #2: No

4. Is the manuscript presented in an intelligible fashion and written in standard English?

Reviewer #1: No

Reviewer #2: Yes

5. Review Comments to the Author

Reviewer #1: The study titled "Diagnostic accuracy of the aortic dissection detection risk score alone or with D-dimer for acute aortic syndromes: Systematic review and meta-analysis" conducted by Ren et al. evaluated the effect of D-dimer on AAS. The study is well-written. I have some major comments:

1- Update your search from inception to January 2024.

2- Define abbreviations in their first use and only use abbreviated forms after definition (e.g., CTA in abstract).

3- The structure of abstract is not appropriate. Use background/methods/results/conclusions headings.

4- Remove subheadings from the discussion.

5- The "strengths and limitations" subheading should be before conclusions.

Reviewer #2: The study is of utmost significance, although certain aspects require clarification.

1- The p-value, AUC, and heterogeneity measures (e.g. I^2) are crucial outcomes of the Reitsma model. It is essential to include these values in the result for all analyses.

2- Certain sections of the manuscript require paraphrasing in order to prevent confusion and ensure the effective conveyance of information, thus necessitating proofreading.

3- What was the author's rationale for excluding trauma patients and incidental patients from being eligible for inclusion in the study?

4- The systematic searches and subsequent findings must be updated, as it has been over a year since the last database search.

5- Authors should ensure that search queries are provided in the supplementary data.

6- The Scottish Intercollegiate Guidelines Network should be cited accurately.

7- Authors should provide detailed explanations on how they justify the combining of various studies with different reference standard tests.

8- In table 3, the authors should provide an explanation for the presence of two columns dedicated to Sensitivity and Specificity in the table's footnote.

9- The discrepancy in the figure representation of the "Summary plot for Canadian guideline based strategy analysis (N=6)" hinders its comparability. Authors should provide an explanation for this decision or modify of the figure.

10- It is recommended to perform a meta-regression analysis considering age and the percentage of females due to the nature of the subject and the potential effects of these covariates.

6. PLOS authors have the option to publish the peer review history of their article (what does this mean?). If published, this will include your full peer review and any attached files.

Reviewer #1: No

Reviewer #2: No

---

## [Author Response · Author response to Decision Letter 0]

5 Apr 2024

We have submitted the response to reviewers as a separate document.

---

## [Decision Letter · Decision Letter 1]

13 May 2024

Diagnostic accuracy of the aortic dissection detection risk score alone or with D-dimer for acute aortic syndromes: Systematic review and meta-analysis

PONE-D-23-42764R1

Dear Dr. Goodacre,

We’re pleased to inform you that your manuscript has been judged scientifically suitable for publication and will be formally accepted for publication once it meets all outstanding technical requirements.

Kind regards,

Amir Hossein Behnoush

Academic Editor

PLOS ONE

Additional Editor Comments (optional):

Reviewers' comments:

Reviewer's Responses to Questions

**Comments to the Author**

1. If the authors have adequately addressed your comments raised in a previous round of review and you feel that this manuscript is now acceptable for publication, you may indicate that here to bypass the “Comments to the Author” section, enter your conflict of interest statement in the “Confidential to Editor” section, and submit your "Accept" recommendation.

Reviewer #1: All comments have been addressed

2. Is the manuscript technically sound, and do the data support the conclusions?

Reviewer #1: (No Response)

3. Has the statistical analysis been performed appropriately and rigorously? 

Reviewer #1: (No Response)

4. Have the authors made all data underlying the findings in their manuscript fully available?

Reviewer #1: (No Response)

5. Is the manuscript presented in an intelligible fashion and written in standard English?

Reviewer #1: (No Response)

6. Review Comments to the Author

Reviewer #1: (No Response)

7. PLOS authors have the option to publish the peer review history of their article (what does this mean?). If published, this will include your full peer review and any attached files.

Reviewer #1: No

---

## [Editor Report · Acceptance letter]

7 Jun 2024

PONE-D-23-42764R1 

PLOS ONE

Dear Dr. Goodacre, 

I'm pleased to inform you that your manuscript has been deemed suitable for publication in PLOS ONE. Congratulations! Your manuscript is now being handed over to our production team.

Kind regards, 

on behalf of

Dr. Amir Hossein Behnoush 

Academic Editor

PLOS ONE